# Acquired syphilis in older people in Brazil from 2010–2020

**Josiane Araújo da Cunha**[1,2]*, **Marquiony Marques dos Santos**[2¤], **Kenio Costa de Lima**[1,3]

**1** Graduate Program in Health Sciences, Federal University of Rio Grande do Norte, Natal, Rio Grande do Norte, Brazil, **2** Laboratory of Technological Innovation in Health, Federal University of Rio Grande do Norte, Natal, Rio Grande do Norte, Brazil, **3** Graduate Program in Collective Health, Federal University of Rio Grande do Norte, Natal, Rio Grande do Norte, Brazil

¤ Current address: Federal University of Rio Grande do Norte, University Hospital Onofre Lopes, Natal, Rio Grande do Norte, Brazil

* josianeac12@gmail.com

**Data Availability Statement:** All relevant research data can be found in the document and the Zenodo data repository (https://doi.org/10.5281/zenodo. 10086131).

## Abstract

### Background

The infection caused by *Treponema pallidum* remains a severe public health problem, with a high prevalence in individuals over 60 years. However, research into infections such as syphilis continues to be neglected in geriatrics. This study aims to evaluate data on the detection rate of syphilis in Brazil, in individuals between 60 and 120 years, by characterizing the epidemiological profile and respective factors associated with it, in addition to performing a temporal trend analysis, from 2010 to 2020.

### Methodology

Ecological, time-series study, which started with the collection of notifications from the database of the Information System on Compulsory Notification of Diseases. The epidemiological profile was characterized based on sociodemographic variables. The Statistical Package for the Social Sciences program, version 19.0, was used for incidence and prevalence analysis. The 2010 census by the Brazilian Institute of Geography and Statistics and projections for older people were considered. In the trend analysis, the *Joinpoint regression model* was used.

### Main findings

There was an upward variation in the detection of syphilis in older people from 2010 to 2018, with a peak in 2018. In 2019, there was a slight reduction in the notification of cases, which was accentuated in 2020. The prevalence of the infection was equivalent to 12.84 cases for each 100,000 Brazilians, with a mean age of 68.04 years (±7.15) for those between 60 and 120 years, being higher in white and black males. The highest proportion of older people with syphilis occurred in the South and Southeast regions. However, the trend analysis showed a significant and homogeneous increase in all regions of Brazil, for both sexes and all age groups.

**Funding:** The Coordination for the Improvement of Higher Education Personnel - Brazil (CAPES) - Finance Code 001 - is financing the payment of publication fees for the PLOS ONE journal. The funder had no involvement in the study design, data collection and analysis, publication decision, or manuscript preparation.

**Competing interests:** The authors have declared that no competing interests exist.

## Conclusions

There is a tendency for an increase in cases of syphilis in older people, which reinforces the need to plan health actions to combat the disease in this population.

## Introduction

The Sexually Transmitted Infection (STI) caused by *Treponema pallidum* [1] remains a significant public health problem, although it was originated in the late 1400s and discovered in 1905 by Schaudinn & Hoffmann [2, 3]. The transmission is mainly through sexual intercourse (oral, vaginal, or anal). Most people with syphilis are asymptomatic, which contributes to maintaining the chain of transmission. However, if untreated, it may cause severe systemic complications after 3–5 years of infection [4].

Syphilis infection is divided into the stages of recent syphilis—primary, secondary, and recent latent, with one year of evolution—and late syphilis, late latent, and tertiary, with more than one year of evolution [5]. The primary stage is characterized by contagion and development of a chancre, usually only one and painless. Signs of secondary syphilis appear between three and eight weeks after the primary chancre. In this stage, cutaneous-mucosal manifestations are predominant, but they usually disappear within a few weeks [6]. The disease continues into the latent phase, without symptoms or signs, but with the risk of developing tertiary syphilis [7]. In the tertiary stage, the nervous and cardiovascular systems are affected [8].

*Treponema pallidum* can be detected in lesions of primary and secondary syphilis. Antibodies against the bacteria are detectable up to 15 days after the appearance of the chancre, and the disease can be diagnosed [9]. Indirect methods for diagnosing this STI have a complex interpretation in the case of reinfected patients since treponemal antibodies remain mostly positive throughout life. Therefore, tests are not as sensitive in early infection or when patients are contagious, and nontreponemal titers are more difficult to interpret after treatment [10].

Syphilis has been described as the great imitator due to its many and varied clinical manifestations. In 2020, 7.1 million new cases of infection by the *Treponema pallidum* were recorded, with a prevalence of 22.3 million. In the American continent, the prevalence in adults, in that same year, for both sexes, was 5,800,000. The incidence rate in adults per 1,000 people, in 2020, for both sexes, was 4.8, with the total number of cases being 2,500,000 [11]. Data indicate a potential increase in new cases of syphilis if adequate interventions to combat its spread do not occur [6].

In the United States, the incidence of syphilis has reached levels not seen in more than 20 years, with an increase in the number of cases reported for both sexes, indicating a growth in the spread of the disease [12, 13]. In China, a study by Tao et al. 2020 [14] found that the increasing proportion of asymptomatic latent syphilis diagnoses probably results from the scale-up of syphilis screening. For Peeling et al. 2017 [15], despite the availability of diagnostic tests and treatment, syphilis is re-emerging as a global public health problem.

From 2012 to 2022, 1,237,027 cases of acquired syphilis were reported in Brazil. Throughout the historical series, the disease detection rate showed an average annual growth of 35.4% until 2018, remaining stable in 2019. In 2020, there was a 23.4% decline in this rate, probably due to reduced diagnostics capacity during the COVID-19 pandemic. In 2021 and 2022, detection rates reached levels higher than the pre-pandemic period, with a 26.6% growth. From 2012 to June 2023, SINAN reported a total of 1,340,090 cases of acquired syphilis, of which 14.2% occurred in the Northeast region. In Rio Grande do Norte, from 2012 to 2022, the

number of cases per 100,000 inhabitants increased from 275 to 2,576. By June 2023, there were already 1,546 cases [16].

As explained by Pagani et al. 2021 [8], an important factor that has changed the characteristics of patients with syphilis is its growth among older people. The increase in life expectancy and quality, in addition to the availability of drugs that allow for prolonged sex life, has reshaped sexual behavior, making these individuals more vulnerable to STIs, such as syphilis. The high prevalence of these infections in individuals over 60 years must be highlighted, including those in extreme age groups, aged 90 years or more [17–20].

On the other hand, sexually transmitted infections such as syphilis remain a neglected area of research in geriatrics and, therefore, further studies are needed on their prevalence in older populations [21]. Since the infection has a variable appearance, other diagnoses may be suggested, implying an underdiagnosis of STIs for older people [22].

In this sense, the relevance of proposing a study which verifies whether the increase in syphilis also occurred among older individuals in a 10-year time frame helps in combating the spread of the disease and outlining perspectives of fighting the disease in this age group, which sometimes is not considered at risk. Older people are usually forgotten when sexually transmitted infections are investigated since structural ageism prevents them from being seen as sexually active.

Therefore, this study aims to evaluate the data on the detection rate of syphilis in Brazil, characterizing the epidemiological profile and respective factors associated with it, in addition to performing a temporal trend analysis, from 2010 to 2020.

## Materials and methods

### Type of study

This is an ecological, time-series study, involving all federative units of Brazil. The Brazilian territory is divided into five regions: North, Northeast, South, Southeast and Central-West, with a total of 203,080,756 inhabitants. Currently, the population aged 60 years or more totals 32,113,490 individuals. The demographic density in the last census survey was estimated at 23.8 inhabitants per square kilometer. Gini index data fell to 0.518 in 2022 and the current HDI is 0.759 [23].

### Data source

Data on acquired syphilis were obtained from a database that receives input from the Notifiable Diseases Information System—SINAN (https://zenodo.org/doi/10.5281/zenodo.10086130) [24]. Therefore, there was no need to submit this research for an ethics review. This system is the main source of information on the notification and registration of cases of syphilis. Therefore, this work uses a national database that contains information that is legitimately associated with the testing/notification of cases of acquired syphilis and its increase in Brazil.

In Brazil, the epidemiological surveillance model comprises investigation and compulsory notification, sentinel services, and cross-sectional studies in population groups. Suspected or confirmed cases on the national list of compulsory notifications are registered at SINAN and follow a flow until the public disclosure of information. In SINAN, only the identification and demographic data of individuals (gender, age, race or skin color, schooling, and residence) are entered. Clinical, laboratory, and behavioral data variables are not included. In addition to SINAN, other systems can be routinely accessed to obtain additional information about STIs. The interoperability of data from municipal, state, and federal epidemiological surveillance services allows the extraction of information on deaths from the Mortality Information System and consultation of Hospital and Outpatient Information Systems, among others [25].

Among the indicators generated through data from the SINAN system, there are those provided by the regular input report in the system and those relevant to the management and innovation of the Unified Health System. Reports are generated from DBF databases of the NET or ONLINE versions of SINAN. This information helps in monitoring disease trends, as well as proposing prevention and control measures [26].

In this sense, SINAN is the main source of secondary data on syphilis notification in Brazil, in which all registered cases of the disease are available. Compulsory notification is mandatory and conducted upon suspicion or confirmation of the disease, being performed by health professionals or any member of the care team who cares for the patient first, within 24 hours. The health authorities that receive the notification must inform the other spheres of SUS management so that it can be recorded and shared among health information systems [27, 28].

To notify cases of acquired syphilis, the SINAN individual investigation/notification form must be used (S1 File https://doi.org/10.5281/zenodo.11638624). In 2017, the criteria for cases of this disease were redefined, in accordance with the guidelines of the Pan American Health Organization: asymptomatic individuals with a reactive non-treponemal test, with any titer, and reactive treponemal test and no record of previous treatment; or symptomatic individuals, with at least one reactive test, treponemal or non-treponemal, with any titer [29].

After the diagnosis of syphilis, the clinical stage is classified according to the time of infection. Patients are treated with benzathine benzylpenicillin, monitored with a non-treponemal test, and the case is reported. If there is a serological scar, patients must be oriented. To control cure, monitoring with a non-treponemal test should occur and, if possible, with the same diagnostic method. During follow-up, non-treponemal tests should be performed every three months until the 12th month of follow-up. The treated person may be released, subject to screening, according to sexual history and risk management for syphilis and other STI [30].

## Study population

The data collection period comprises the beginning of the actual notification of acquired syphilis in Brazil, which became mandatory in 2010. The study was based on information collected from the official public health database system of the Brazilian Ministry of Health—The Notifiable Diseases Information System—SINAN. It included 71,289 older individuals, aged between 60–120 years old.

## Variables

The information regarding all these variables was obtained from the SINAN database. The main variable considered for characterizing the epidemiological profile was the diagnosis of acquired syphilis in those aged 60 years or more. The independent variables corresponded to those of sociodemographic interest: the reporting of the first symptom onset; age group (60–64, 65–69, 70–74, 75–79, 80–84, 85–89, and 90 or more); gender (male and female); race/color (white, yellow, indigenous, and black); level of education (illiterate, primary, secondary, and higher education); region of the country in which the federative units are grouped; and classification of syphilis according to the cure or death statuses.

Descriptive analyses were performed, with the qualitative variables of interest expressed in percentage values. Missing data and information not reported by respondents were considered for all variables.

## Analysis of the detection rate of acquired syphilis

The detection rate was calculated considering the 2010 census by the Brazilian Institute of Geography and Statistics (IBGE) and the projections for the Brazilian older population

(individuals aged 60 years or more), from 2010 to 2020 [31]. Data analysis was based on the Statistical Package for the Social Sciences—SPSS®, version 19.0. The detection rate was obtained by the number of cases of syphilis divided by the number of older people in each period, posteriorly multiplied by a constant– 100,000.

## Trend analysis

Based on the annual rates of acquired syphilis, a regression model based on inflection points was used (*Joinpoint regression model*) for analyzing temporal trends. This method verifies whether a line with multiple segments is statistically better to describe the temporal evolution of a data set when compared to a straight line or one with fewer segments. The model also allows calculating the trend of the *Annual Percent Change*–APC and *Average Annual Percent Change*–AAPC indicators, that is, the analysis of the acquired syphilis trend per year and throughout the period. The AAPC was categorized as increasing (APC/AAPC+ and $p < 0.05$), stable (p-value $> 0.05$), or decreasing (APC/AAPC- and $p < 0.05$), which is an important definition to understand the factors that influence the variation trend regarding syphilis [32–34].

The *Joinpoint Trend Analysis Software* identified moments (years) in which there was a significant change in the detection rate of this STI, observed at different points in time (inflection points). In each segment, the APC was estimated as a descriptive measure of the changes in time trends. A Confidence Interval of 95% (95% CI) and a significance level of 5% ($p < 0.05$) were considered [35]. The *Microsoft Excel Office 2013* software was used to organize the databank, build indicators, and prepare tables.

## Results

### Analysis of the detection rate of acquired syphilis

This study sought to characterize, in the most reliable way, the epidemiological profile of older people infected with acquired syphilis who are users of the public health network.

When analyzing the detection rate of syphilis in Brazil, between 2010 and 2020, a progressive increase in this STI has been observed in older people—aged between 60 and 120 years—since 2011, which intensified in 2018 (Fig 1). Among the notified cases, the detection rate was equivalent to 12.84 cases for every 100,000 Brazilians, with 34.74 cases per 100 thousand individuals for males and 18.46 cases per 100 thousand individuals for females. About 38 older individuals (0.05%) presented no information regarding gender.

In this analysis, white, black, yellow, and indigenous older individuals presented the following detection rates per 100,000 people: 5.45; 5.11; 0.10, and 0.07 cases. For this variable, there was an abstention of 11,066 (15.50%) responses and 599 (0.84%) cases of missing data.

Regarding their level of education, 3,702 (5.19%) individuals were illiterate; 5,016 (7.04%) revealed having primary education; and 5,150 (7.22%) had high school education. On the other hand, only 1,480 (2.08%) older individuals had higher education. Of the total information, 28,037 (39.33%) individuals had their data on education level considered lost.

In this study, regarding the regions of Brazil, it is highlighted that the highest rates of syphilis detection in older individuals occurred in the South and Southeast (even higher than the national rate), and the lowest in the Northeast. In this sense, it is essential to emphasize the high detection rate of syphilis in older people when considering the number of existing cases in the total older population (Fig 2).

As for the age groups most diagnosed with syphilis, there was a predominance of those between 60–64 and 65–69 years (Fig 3). The mean age was 68.04 years (±7.15).

Regarding the clinical evolution of STIs in older people (Fig 4), of the total number of cases, approximately 33,322 individuals were cured between 2010 and 2020 (detection rate of 123.33

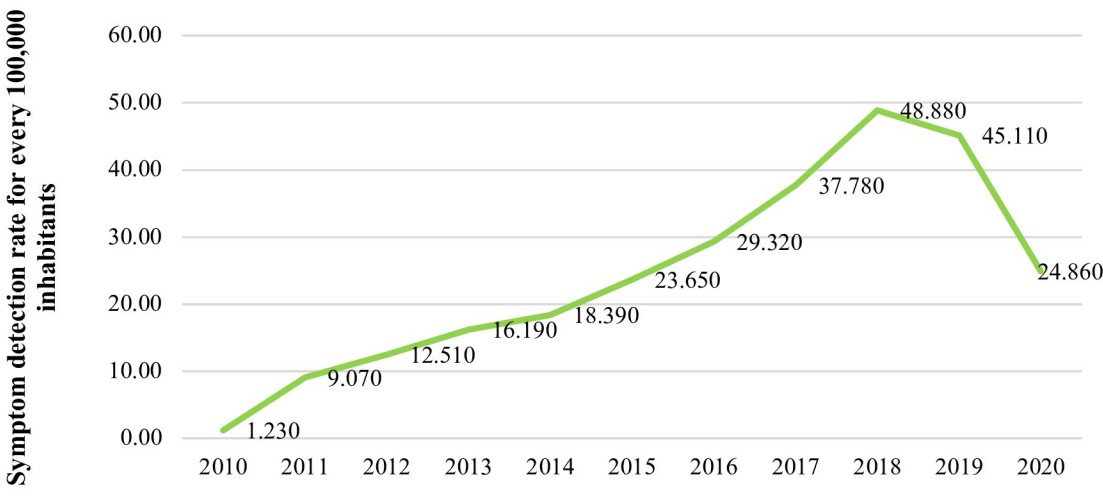

**Fig 1. Detection rate of acquired syphilis, according to the year of reporting the first symptoms, from 2010 to 2020.** https://doi.org/10.5281/zenodo.11638624.

infected in every 100,000 older people in Brazil). As there are specific cases of patients with syphilis whose investigation was closed in 2021–892 cases out of 71,289—it is noteworthy that 344 individuals were cured in 2021 and eight died, two due to acquired syphilis and six due to other causes. In that year, for 148 cases, information on cure or death was ignored and for 392 individuals the information was absent.

The highest cure rates occurred from 2014 to 2018, with a slight decrease in 2019. Of 255 deaths, 0.94 in every 100,000 older people occurred from 2010 to 2020, with 16.86% of them being due to syphilis infection and 83.14% due to other causes.

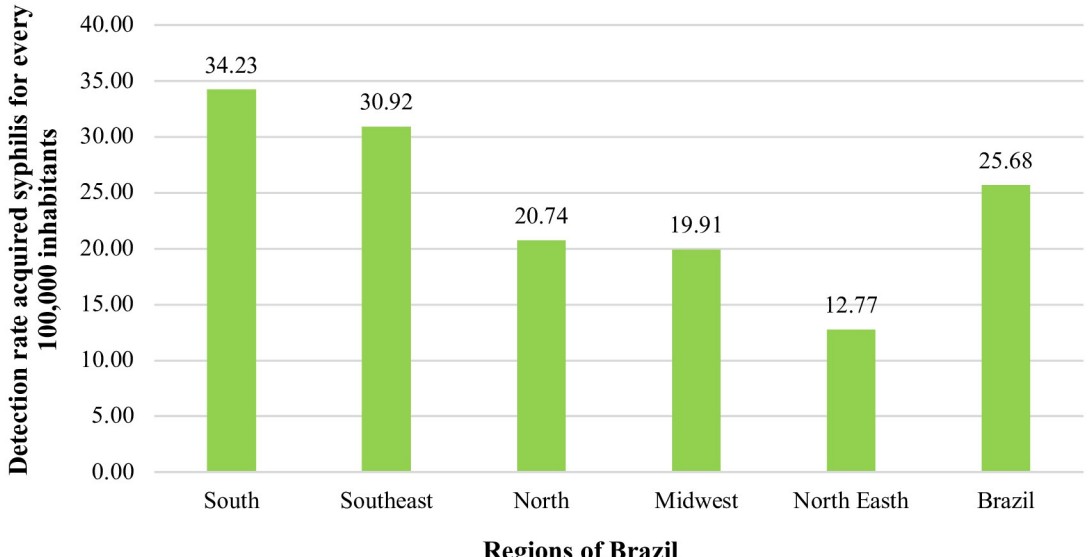

**Fig 2. Detection rate of acquired syphilis in older people, according to regions of Brazil, from 2010 to 2020.** https://doi.org/10.5281/zenodo.11638624.

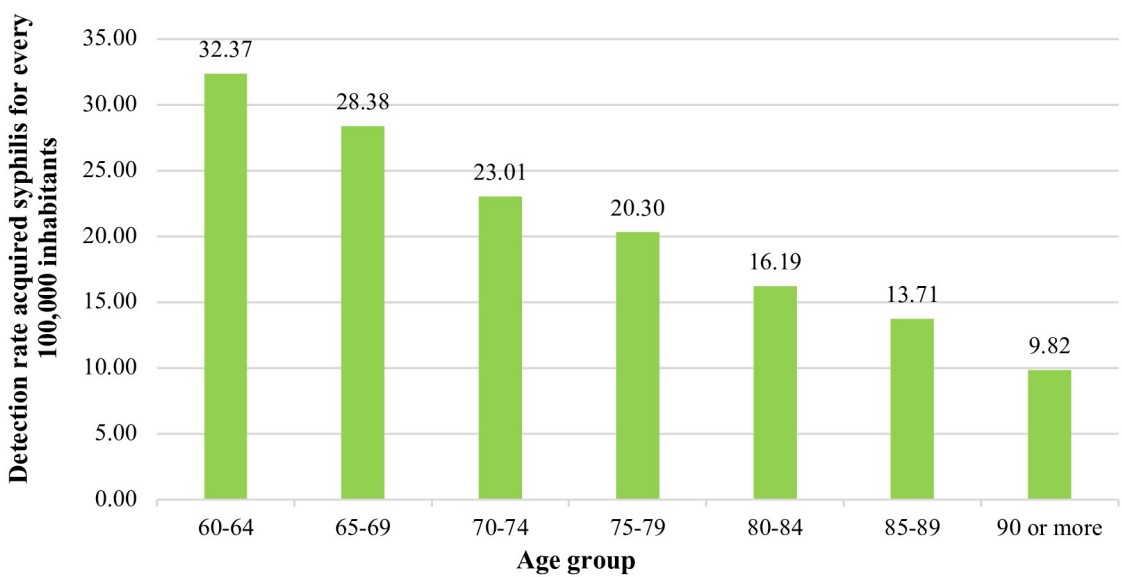

**Fig 3. Classification of older people, according to the detection rate of acquired syphilis by age group, from 2010 to 2020.** https://doi.org/10.5281/zenodo.11638624.

Notably, from 2010 to 2020, a representative rate, equivalent to 23.31% of cases, refers to the code "ignored". It should be noted that, of 69,063 older people with syphilis in this period, 28.07% did not present information regarding the conclusion of the disease. For 1,334 individuals, that is, 1.87% of cases, there was no information on the year in which the disease ended.

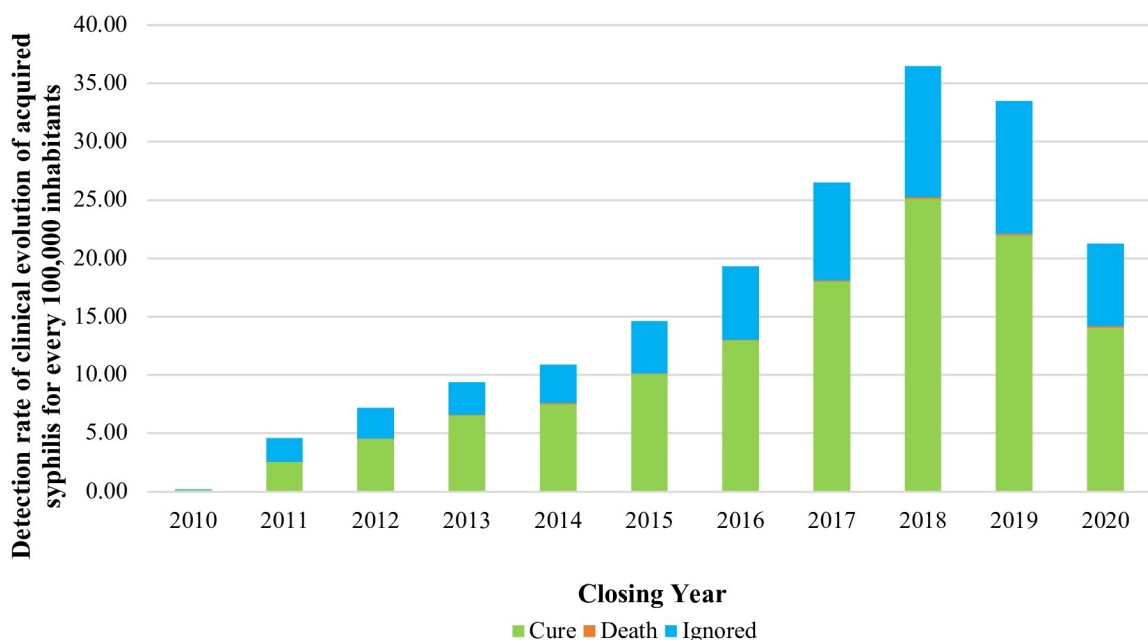

**Fig 4. Detection rate of the clinical evolution of acquired syphilis in older people, according to the year of completion of the investigation.** https://doi.org/10.5281/zenodo.11638624.

## Trend analysis

From the calculation of the annual percentage change (APC) and average annual percentage change (AAPC) of syphilis rates in older people, by region of Brazil and according to gender and three age groups (60–69 years; 70–79 years and 80 years or more), it was found that the growth trends of the disease were significantly homogeneous, considering the variables mentioned (Table 1).

Given the data presented, it was observed that there was a significant increase in the trend of syphilis cases for males and all age groups in the Northeast region until the year 2020. As for females, in the age groups 60–69 and 80 years or more, the increasing annual percentage change was significant, especially up to 2018. For the 70–79 age group, the APC and AAPC were equivalently increasing.

Considering males in the 60–69 age group in the North region, there are two increasing and significant annual percentage variation cut-off points—from 2010–2013, in which the increase was more pronounced, and from 2013–2020. The APC and AAPC are equivalently increasing and essential for the other age groups. Regarding females, there was a significant increase in syphilis rates for the 60–69 age group, starting in 2012. For the other age groups, the APC and AAPC follow the same pattern as for males.

Regarding the Center-West region, for males, the age group of 60–69 years showed a significant increase until 2018. In the succeeding age groups, the APC and AAPC increased equivalently and significantly up to 2020. For females, a significant increase in cases occurred in all age groups until 2020 (APC = AAPC). In the Southeast region, for all age groups and both genders, there was a significant annual percentage variation increasing significantly until 2012, remaining practically constant, especially for females.

In the South region, for males aged 60–69 and 70–79, the annual percentage variation significantly increased from 2010 to 2012 and from 2012 to 2020. The increase was significant since 2012, for those with 80 years or more. As for females, in the 60–69 age group, the increasing annual percentage variation was significant from 2010 to 2013 and 2013 to 2020. The increase was significant for the 70–79 age group, from 2010 to 2015. For those aged 80 years or more, the increase was significant from 2010 to 2016.

## Discussion

This study contributes to the literature by demonstrating the impact of acquired syphilis on older individuals in Brazil.

When analyzing the detection rate of the disease, from 2010 to 2020, there was a progressive increase in STIs in older people, mainly in the years 2016 and 2017, with a peak in 2018. Despite the obligation, as of 2010, of health units to report cases of syphilis, which reflects a significant magnitude of the disease, data from the country still indicate underreporting. This compromises health actions aimed mainly at older people [36].

The underreporting of cases in the Notifiable Diseases Information System has relevant implications for the response to STIs in the country, as important information involving behaviors and vulnerabilities remains unknown. Furthermore, the lack of registration may compromise priority actions to combat the disease. In this sense, it is essential that there is timely notification of all cases, as well as continuous improvement in the quality of filling out the notification and investigation form [30].

The study highlighted the increase in cases of syphilis in older people. It is noteworthy that, in recent years, this increase has been observed in all age groups. This can be attributed, in part, to high-risk sexual activity with the reduction in the use of condoms and the

**Table 1. Trends of syphilis in older people, according to the region and gender of the individual, in Brazil.**

| Sex | Age Group | Period | APC | AAPC | p-value | CI 95% | Trend |
|---|---|---|---|---|---|---|---|
| | | | | Northeast Region | | | |
| Male | 60 to 69 | 2010–2020 | - | 50.5* | <0.001 | 31.7–72.1 | Increasing |
| | 70 to 79 | 2010–2020 | - | 47.9* | <0.001 | 32.9–64.7 | Increasing |
| | 80 or more | 2010–2020 | - | 47.5* | <0.001 | 32.1–64.8 | Increasing |
| Female | 60 to 69 | 2010–2020 | - | 43.4* | | 24.0–65.7 | Increasing |
| | | 2010–2018 | 70.3* | - | <0.001 | 55.6–86.3 | Increasing |
| | | 2018–2020 | -28.0 | - | | -68.5–64.8 | Stable |
| | 70 to 79 | 2010–2020 | | 51.6* | <0.001 | 32.7–73.1 | Increasing |
| | 80 or more | 2010–2020 | | 35.3* | | 14.1–60.4 | Increasing |
| | | 2010–2018 | 60.9* | - | <0.001 | 44.7–78.9 | Increasing |
| | | 2018–2020 | -32.3 | - | | -74.4–79.1 | Stable |
| | | | | North Region | | | |
| Male | 60 to 69 | 2010–2020 | - | 57.6* | | 38.0–79.9 | Increasing |
| | | 2010–2013 | 125.5* | - | 0.005 | 41.3–259.9 | Increasing |
| | | 2013–2020 | 35.1* | - | 0.001 | 19.3–53.1 | Increasing |
| | 70 to 79 | 2010–2020 | - | 47.5* | <0.001 | 32.6–64.1 | Increasing |
| | 80 or more | 2010–2020 | - | 52.1* | 0.001 | 24.9–85.2 | Increasing |
| Female | 60 to 69 | 2010–2020 | - | 54.8* | | 26.9–88.8 | Increasing |
| | | 2010–2012 | 170.5 | - | | -13.3–743.3 | Stable |
| | | 2012–2020 | 34.6* | - | 0.001 | 18.9–52.4 | Increasing |
| | 70 to 79 | 2010–2020 | - | 49.2* | <0.001 | 36.6–63.0 | Increasing |
| | 80 or more | 2010–2020 | - | 33.6* | 0.001 | 15.8–54.0 | Increasing |
| | | | | Center-West Region | | | |
| Male | 60 to 69 | 2010–2020 | - | 26.7* | | 14.9–39.8 | Increasing |
| | | 2010–2018 | 38.6* | - | <0.001 | 30.3–47.3 | Increasing |
| | | 2018–2020 | -11.4 | - | | -49.4–55.3 | Stable |
| | 70 to 79 | 2010–2020 | - | 31.6* | <0.001 | 22.3–45.1 | Increasing |
| | 80 or more | 2010–2020 | - | 36.4* | <0.001 | 21.6–53.1 | Increasing |
| Female | 60 to 69 | 2010–2020 | - | 29.4* | <0.001 | 20.1–39.5 | Increasing |
| | 70 to 79 | 2010–2020 | - | 28.2* | <0.001 | 18.0–39.3 | Increasing |
| | 80 or more | 2010–2020 | - | 37.5* | 0.002 | 17.0–61.6 | Increasing |
| | | | | Southeast Region | | | |
| Male | 60 a 69 | 2010–2020 | - | 38.7* | | 19.1–61.6 | Increasing |
| | | 2010–2012 | 329.1* | - | 0.006 | 79.3–926.5 | Increasing |
| | | 2012–2020 | 4.6 | - | | -4.9–15.1 | Stable |
| | 70 a 79 | 2010–2020 | - | 43.6* | | 23.2–67.5 | Increasing |
| | | 2010–2012 | 401.6* | - | 0.004 | 108.3–1107.8 | Increasing |
| | | 2012–2020 | 5.1 | - | | -4.6–15.6 | Stable |
| | 80 or more | 2010–2020 | | 27.5* | | 10.2–47.6 | Increasing |
| | | 2010–2012 | 144.6* | - | 0.04 | 5.9–465.1 | Increasing |
| | | 2010–2020 | 8.4 | - | | -1.1–18.7 | Stable |

(*Continued*)

**Table 1.** (Continued)

| Sex | Age Group | Period | APC | AAPC | p-value | CI 95% | Trend |
|---|---|---|---|---|---|---|---|
| Female | 60 a 69 | 2010–2020 | - | 38.9* | | 15.2–67.4 | Increasing |
| | | 2010–2012 | 385.6* | | 0.011 | 67.0–1312.1 | Increasing |
| | | 2012–2020 | 1.6 | | | -9.6–14.1 | Stable |
| | 70 a 79 | 2010–2020 | | 38.2* | | 14.9–66.2 | Increasing |
| | | 2010–2012 | 326.9* | | 0.015 | 48.6–1126.5 | Increasing |
| | | 2012–2020 | 4.2 | | | -7.1–16.9 | Stable |
| | 80 or more | 2010–2020 | | 36.5* | | 13.3–64.5 | Increasing |
| | | 2010–2012 | 355.0* | | 0.013 | 56.5–1222.8 | Increasing |
| | | 2012–2020 | 1.0 | | | -10.1–13.5 | Stable |
| South Region | | | | | | | |
| Male | 60 a 69 | 2010–2020 | - | 91.5* | | 59.0–130.7 | Increasing |
| | | 2010–2012 | 993.0* | - | 0.002 | 277.4–3065.8 | Increasing |
| | | 2012–2020 | 23.9* | - | 0.004 | 10.3–39.2 | Increasing |
| | 70 a 79 | 2010–2020 | - | 67.4* | | 40.4–99.7 | Increasing |
| | | 2010–2012 | 370.3* | - | 0.009 | 71.7–1188.2 | Increasing |
| | | 2012–2020 | 29.3* | - | 0.001 | 15.9–44.3 | Increasing |
| | 80 or more | 2010–2020 | | 62.7* | | 23.8–113.7 | Increasing |
| | | 2010–2012 | 281.8 | | | -20.0–1721.6 | Stable |
| | | 2012–2020 | 31.4* | | 0.008 | 10.8–55.8 | |
| Female | 60 a 69 | 2010–2020 | | 67.7* | | 42.6–97.3 | Increasing |
| | | 2010–2013 | 239.2* | | 0.002 | 91.3–501.6 | Increasing |
| | | 2013–2020 | 24.0* | | 0.014 | 6.4–44.5 | Increasing |
| | 70 a 79 | 2010–2020 | | 56.2* | | 32.3–84.5 | Increasing |
| | | 2010–2015 | 113.2* | | 0.001 | 58.9–186.0 | Increasing |
| | | 2015–2020 | 14.5 | | | -14.7–53.6 | Stable |
| | 80 or more | 2010–2020 | | 54.5* | | 26.3–89.1 | Increasing |
| | | 2010–2016 | 92.5* | | 0.001 | 48.1–150.4 | Increasing |
| | | 2016–2020 | 11.1 | | | -32.0–81.7 | Stable |

* Statistical significance

Results of trend analysis by regions of Brazil

improvement of the epidemiological surveillance system, which allows for increasingly effective reporting of data [30].

Although the growing trend may be caused by an increase in reporting, it is common knowledge that acquired syphilis has increased worldwide, including in Latin America, and has re-emerged in several regions, such as North America, Western Europe, China, and Australia [37]. In this context, compulsory notifications cannot be pointed out as the isolated reason for the increase in cases.

In the years 2018, 2019, 2020, 2021, 2022, in Brazil, 158,051, 152,915; 115,371; 167,523, 213,129 cases of acquired syphilis were reported, respectively. During 2020, there was a considerable reduction in the number of notifications [16, 38–41], probably due to the particularity of the COVID-19 pandemic, caused by SARS-CoV-2 [42]. Between 2018 and 2019, and 2021 and 2022, there was a 22.41% increase in notifications of acquired syphilis in Brazil [16, 38–41].

On the same premise, the pre-pandemic period, from 2012 to 2018, revealed a growing trend in the detection rates of acquired syphilis at all ages, with a greater number of cases as the age group decreased: 41.6% among individuals of 13 to 19 years old; 39.2% among those aged 20 to 29; 30.8% between those aged 30 and 39; 26.2% in those aged 40 to 49, and 25.4% in those aged 50 or over. Then, there was some stability in 2019 and a reduction in 2020. In the post-pandemic period, between 2021 and 2022, there was an increase in reported cases for all age groups, the percentage of which remained higher in individuals aged 40 to 49 and 50 or over [16].

Regarding the detection rate according to sex, the research found a more significant number of cases in males, probably because men less frequently seek specialized medical care [43, 44]. The current Syphilis Epidemiological Bulletin [16] showed that, between 2012 and 2022, 750,848 (42.3%) cases of acquired syphilis occurred in men and 485,115 (47.4%) in women.

Corroborating these findings, Barros et al. 2018 [45] detected a high prevalence of STIs among homeless men. Likewise, a study on syphilis acquired in a Reference Center on STIs and AIDS, in São Paulo/Brazil, observed that most cases were in males [46]. A global trend analysis, from 1990 to 2019, identified that the incidence rates of syphilis and other STIs increased considerably in men [47]. Research involving patients with neurosyphilis (aged 17–75 years) revealed that most of the infected were males [48].

Pinto et al. 2018 [49] when evaluating the prevalence and factors associated with HIV infection and syphilis, in men and women, between 15 and 64 years old, noted that the majority of cases were also in men. Bourchier et al. 2020 [50] when evaluating the incidence of STIs in women, found that the increase occurred mainly in older women, aged between 55 and 74 years. Yeganeh et al 2021 [51] reinforced that sexually transmitted infections in women can originate from the male sexual partners, who remains underdiagnosed and untreated due to lack of symptoms and decreased access to health care due to not seeking health services.

Most of the individuals analyzed in this research were white and black. When comparing notifications in older individuals from 2012 to 2022, the increase in the percentage of syphilis cases according to race/color was 50.7% for black and brown individuals [16]. Smock et al 2017 [52], when analyzing syphilis infections from 2001 to 2013, found that the incidence rate increased in all groups, especially among black individuals [53], when investigating the prevalence of syphilis in individuals aged 18–64 years, found that the majority were black. These circumstances highlight how interventions to combat STIs should be oriented, considering marginalized populations.

Most of the sample in this research had primary or secondary education. It is known that a good level of education is one of the key factors for understanding preventive measures against diseases such as syphilis [17, 54]. On the other hand, having a lower level of education is one of the risk factors for acquiring STIs [36, 54]. Individuals with a low educational level often have little or no access to health-related benefits such as educational programs [55].

In this study, older adults, in the 60–64 and 65–69 age groups, had more diagnoses of syphilis, presumably because they were more sexually active. Oliveira e Juskevicius, 2019 [56], in an analysis of HIV and STI epidemiological bulletins found an increase in the number of cases of acquired syphilis in older people.

From the same perspective, a trend analysis of syphilis detection in elderly people found an increase in the detection rate of this STI in individuals aged ≥ 60 years [18]. Takahashi et al. 2022 [21] when evaluating the national trend in the incidence of syphilis in Japan, in patients aged ≥ 50 years, identified a significant increase of this STI in these individuals.

In this study, the data showed higher cure rates for syphilis when observing the clinical evolution of patients with STIs. On the other hand, for some individuals, there is no information available on the outcome of the disease (cure or death), possibly due to flaws in the records of

patients with suspected infection. In contrast, a comparative analysis developed by Barragan et al. 2017 [57] showed that older people aged 75 to 84 years and ≥ 85 years had the highest frequency of deaths from syphilis.

This study found an increasing trend in the detection rate of syphilis in older people in Brazil, from 2010 to 2020. Regarding the distribution of the disease in the Brazilian territory, although the South and Southeast regions concentrate the highest detection rates of syphilis in older people, growing trends of STIs in these individuals are pointed out, which are also significantly homogeneous (p-value in Table 1) in the five regions of the country, for both sexes and age groups of the population.

The increase in the detection rate highlights the need for adequate planning and development of actions to prevent and combat the disease for this population. The aging of the Brazilian population may be a predictor for an increasing number of syphilis cases in older people, while greater testing is expected in this group, which is increasingly sexually active.

Estimates of the acquired syphilis trend highlight the need for a solid approach to disease control in the country. Therefore, it is essential to expand the qualification of health actions, in order to provide adequate conditions to combat the disease [58].

Corroborating our findings, Barros et al. 2023 [18] also found an increasing trend in the detection rate of syphilis in older people in Brazil, from 2011 to 2019, for all age groups and both sexes. Wang et al. 2021 [59] identified a significant increase in the incidence rate of syphilis in older people in Guangdong (China), greater than in young adults, from 2013 to 2018. They also observed a trend toward a significant increase in cases of tertiary syphilis (a substantial correlation between the late diagnosis of this STI and advanced age).

A decade-long trend analysis of acquired syphilis in Brazil, considering the different regions of the country, helped to visualize a significant growth in the disease notification rates [60]. Expanding access to rapid syphilis tests has increased its detection rate and, consequently, effective notification. On the other hand, the increase in testing coverage and reorganization of health units did not stabilize the upward curve of the disease [61].

In Brazil, strengthening the Family Health Strategy improved access to health. Among the factors that may explain the increase in the notification of syphilis cases in the country, the acquisition of rapid tests; national prevention campaigns; and the instrumentation of situation rooms in municipalities stand out [62, 63]. However, even with improved notification and high detection of cases, syphilis is still underreported [36].

On the other hand, improving the epidemiological surveillance system and bringing the professionals responsible for data input up to date helps to eliminate under-reporting. Furthermore, the interoperability of information systems enables the identification of the probable population covered by the geographic census, the groups affected by syphilis, and the resulting deaths [30].

The spread of the disease is believed to be driven by late diagnosis; discontinuity in treatment; and discrimination related to targeted sexuality education. With advancing age, it is observed that concerns about health issues are restricted to specific weaknesses related to aging [58, 64, 65]. The most common practice is to investigate neurodegenerative diseases and/or comorbidities, such as Chronic Non-Communicable Diseases. In this sense, public policies to contain the spread of syphilis among individuals over 60 years of age are still scarce [66].

We emphasize that the increase in syphilis rates in younger individuals today will reflect, in the future, on management problems for controlling cases of this STI in older people, a problem that is already observed when evaluating its current detection rates. If the problem is not seriously resolved for all age groups, the tendency is for syphilis to remain a national epidemic for a long time and a permanent concern for health managers.

It is important to emphasize that the increase in life expectancy and quality of life causes behavioral changes in older people. Unlike previous generations, old age is no longer linked to asexuality. The increase in STI rates reflects different practices, such as unprotected sex, probably due to limited knowledge. These individuals do not see themselves as belonging to risk groups [50] and therefore require greater attention from health teams [8].

In this regard, acknowledging change for preventing acquired syphilis and promoting health is expected. It is essential to reorganize services; train Primary Health Care professionals to develop intervention programs aimed at older people, thinking about specific approaches to their education and sexual orientation; and encourage the use of the rapid treponemal tests to detect syphilis and provide early treatment [67].

As for the limitations of this study, difficulties were encountered in data analysis due to a lack of records or underreporting of acquired syphilis. The gaps related to some variables are highlighted, and the absence of information such as disease stages and treatment. In addition, SINAN does not always update the data collected by the notification/investigation forms promptly. Such gaps indicate that the underreporting of the disease may be associated with incorrect completion of the notification forms [55]. However, since this study used national databases; this is the information available that comes closest to demonstrating the increase in acquired syphilis in older people, in Brazil.

## Conclusion

Although the South and Southeast regions concentrate the highest detection rates of syphilis in older people, the temporal analysis research demonstrated a significant increase in the growth trends of syphilis in all five regions of Brazil, affecting both sexes, but mainly white and black individuals and those with low education.

It is observed that inequalities in syphilis care are directly related to individual socioeconomic and demographic characteristics. The increase in the detection rate of this STI among older people is reflected in changes in sexual behavior and highlights the need for targeted health promotion for them, such as guidance on safe sexual practices and commitment to early investigation, diagnosis, and treatment.

In this way, the trend of significant increase in syphilis cases in older people is reaffirmed, reinforcing the need for intersectoral planning of health surveillance actions, with the aim of engaging Primary Care to combat the spread of the disease, especially for this population.

## Supporting information

**S1 File.** https://doi.org/10.5281/zenodo.11638624.
(PDF)

## Acknowledgments

We want to thank the Health Ministry for providing the data used in the confection of the manuscript.

## Author Contributions

**Conceptualization:** Josiane Araújo da Cunha, Kenio Costa de Lima.

**Data curation:** Josiane Araújo da Cunha.

**Formal analysis:** Josiane Araújo da Cunha, Marquiony Marques dos Santos.

**Funding acquisition:** Josiane Araújo da Cunha.

**Investigation:** Josiane Araújo da Cunha, Kenio Costa de Lima.

**Methodology:** Josiane Araújo da Cunha, Kenio Costa de Lima.

**Resources:** Josiane Araújo da Cunha, Kenio Costa de Lima.

**Supervision:** Josiane Araújo da Cunha, Kenio Costa de Lima.

**Validation:** Josiane Araújo da Cunha, Kenio Costa de Lima.

**Visualization:** Josiane Araújo da Cunha, Kenio Costa de Lima.

**Writing – original draft:** Josiane Araújo da Cunha.

**Writing – review & editing:** Kenio Costa de Lima.

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
