## [Decision Letter · Decision Letter 0]

27 Mar 2023

PONE-D-23-02461ACQUIRED SYPHILIS IN AGED PEOPLE IN BRAZIL FROM 2010-2020PLOS ONE

Dear Dr. Cunha,

Thank you for submitting your manuscript to PLOS ONE. After careful consideration, we feel that it has merit but does not fully meet PLOS ONE’s publication criteria as it currently stands. Therefore, we invite you to submit a revised version of the manuscript that addresses the points raised during the review process.

The reviewers have provided their comments for the improvement of the manuscript. They do raise some further points that I recommend are considered in any subsequent revision. These six key points that would need to be addressed can be summarized as:

Revise the introduction to provide a clear overview of syphilis epidemiology among older adults in Brazil and emphasize the study’s significance.Clarify definitions of "incident" and "prevalent" cases in the study, considering diagnostic challenges.Improve the discussion and conclusion based on reviewers' comments.Adhere to appropriate reporting guidelines and include the checklist as a supplementary file.Ensure all data underlying the findings are fully available.Improve the article's clarity and readability through English proofreading.

We look forward to receiving your revised manuscript.

Kind regards,

Delfina Fernandes Hlashwayo, M.Sc.

Academic Editor

PLOS ONE

Reviewers' comments:

Reviewer's Responses to Questions

**Comments to the Author**

1. Is the manuscript technically sound, and do the data support the conclusions?

Reviewer #1: Partly

Reviewer #2: Yes

2. Has the statistical analysis been performed appropriately and rigorously? 

Reviewer #1: Yes

Reviewer #2: Yes

3. Have the authors made all data underlying the findings in their manuscript fully available?

Reviewer #1: No

Reviewer #2: No

4. Is the manuscript presented in an intelligible fashion and written in standard English?

Reviewer #1: Yes

Reviewer #2: No

5. Review Comments to the Author

Reviewer #1: The authors used epidemiological data on syphilis among older adults in order to investigate trends in syphilis incidence and prevalence among older adults in Brazil. Several critical questions remain unanswered based on the data presented.

1. Is it possible that the changes in reporting of syphilis were at least partially accounted for by the fact that the reporting system was implemented in 2010? The authors need to address the question of whether all health reporting systems were online and functional at the same point in time or whether rate increases might have been driven by the newness of the reporting system and the time required for it to be fully functional with a high compliance with reporting.

2. The authors present population based rates (diagnoses / population in the age range of interest), suggesting that rates have increased over time. However, this approach does not account for changes in clinical testing. Given that syphilis may be present and asymptomatic for decades, it is important to understand whether an increase in rates simply reflects more awareness of the issue and more testing among older adults.

3. The authors need to clarify their use of the terms incident cases and prevalent cases. Incident cases should refer to those who are newly infected. However, in the case of syphilis, an infection that is usually diagnosed serologically and can be clinically silent for decades, the only way to be certain the case truly represents a recent infection is based on having primary or secondary clinical symptoms or serial serologies that demonstrate a recent rise in titer on a nontreponemal test. Was this information available from the data source? Likewise, once syphilis is diagnosed, it should be treated and the infection should resolve. It is assumed that the use of the terms "prevalent" is being applied to cases of newly diagnosed syphilis that lack primary or secondary symptoms or were not previously tested with a demonstrated change in titer over time. The usual terminology for these would be latent or late stage syphilis. Clarification is important in order to understand these findings in the context of other literature on syphilis.

Reviewer #2: The article describes a 10 year prevalence, incidence, and trend of syphilis in people aged 60 years and above in Brazil. A 10 year trend analysis showed a significant increase in cases of syphilis in the entire region of Brazil. The study appears to be statistically sound. There is further scope for improvement. Here are my constructive comments:

Major issues

1.The introduction section does not provide background and information relevant to the study. It doesn’t reflect the significance of the study. It should be re-written to emphasize the importance of describing the epidemiology of syphilis among elderly people in Brazil

2.The discussion section needs major improvement.

A.The authors should focus on the most important findings in this study and present it in a well-organized manner.

B.The discussion should be presented in a well organized manner. For example, page 14, line number 280, discusses about prevalence according to gender and again page 16, line number 313-316, discusses about incidence of syphilis among women. This are same idea presented separately but can be discussed under one paragraph . the same applies for the discussion on trend analysis, page 16-17, line number 337-359 and discussion on prevalence and incidence on page 14, line number 260-268

C.Please compare your findings with the findings of other local studies as much as possible.

D.Avoid unnecessary sentences that doesn’t add any value to the discussion. For example, the sentence stated on line number 391-408, page 19, have no relevance to your discussion.

3.The conclusion is lengthy and doesn’t precisely summarize the findings of the study. The conclusion should be summarize the study findings clearly and precisely.

4.While the study appears to be sound, the language is unclear, making it difficult to follow. I advise the authors to work with a writing coach or copyeditor to improve the flow and readability of the text.

Minor issues

1.Substitute “observational, times series study” on page 2, line number 36 with “cross-sectional, time series study”

2.Please follow the STROBE guideline for reporting cross-sectional study and provide the cehck list as supplementary file

6. PLOS authors have the option to publish the peer review history of their article (what does this mean?). If published, this will include your full peer review and any attached files.

Reviewer #1: No

Reviewer #2: No

---

## [Author Response · Author response to Decision Letter 0]

17 May 2023

We appreciate the opportunity to improve this study and hope that we have responded to the requests of the editors and reviewers.

---

## [Decision Letter · Decision Letter 1]

16 Jun 2023

PONE-D-23-02461R1ACQUIRED SYPHILIS IN OLDER PEOPLE IN BRAZIL FROM 2010-2020PLOS ONE

Dear Dr. Cunha,

Thank you for submitting your manuscript to PLOS ONE. After careful consideration, we feel that it has merit but does not fully meet PLOS ONE’s publication criteria as it currently stands. Therefore, we invite you to submit a revised version of the manuscript that addresses the points raised during the review process.

We look forward to receiving your revised manuscript.

Kind regards,

Delfina Fernandes Hlashwayo, M.Sc.

Academic Editor

PLOS ONE

Reviewers' comments:

Reviewer's Responses to Questions

**Comments to the Author**

1. If the authors have adequately addressed your comments raised in a previous round of review and you feel that this manuscript is now acceptable for publication, you may indicate that here to bypass the “Comments to the Author” section, enter your conflict of interest statement in the “Confidential to Editor” section, and submit your "Accept" recommendation.

Reviewer #1: (No Response)

Reviewer #3: (No Response)

Reviewer #4: (No Response)

2. Is the manuscript technically sound, and do the data support the conclusions?

Reviewer #1: Partly

Reviewer #3: No

Reviewer #4: Partly

3. Has the statistical analysis been performed appropriately and rigorously? 

Reviewer #1: I Don't Know

Reviewer #3: Yes

Reviewer #4: Yes

4. Have the authors made all data underlying the findings in their manuscript fully available?

Reviewer #1: (No Response)

Reviewer #3: Yes

Reviewer #4: No

5. Is the manuscript presented in an intelligible fashion and written in standard English?

Reviewer #1: Yes

Reviewer #3: No

Reviewer #4: Yes

6. Review Comments to the Author

Reviewer #1: The authors have adequately addressed prior comments, significantly improving on the manuscript. It is noted that there are still a few small inconsistencies (not noted on prior review) or questions that should be corrected before the manuscript is published.

1. Discussion of low level of education as related to syphilis (357). It is not clearly that low education is causally related to syphilis or even truly associated. In order to study an association, one would have to have a comparison group and show that people with syphilis have lower educational achievement than population norms for this age in Brazil. In addition, the discussion about education has a somewhat pejorative tone "lower education level is associated with lower health concerns...."

2. The conclusion (432) notes black race as associated with syphilis whereas the data (191) demonstrated slightly higher diagnoses in whites (though strikingly similar rates in white and black populations.

3. Proof carefully for pejorative language (334) such as "men tend to be negligent with their health."

4. Finally, when discussing deaths due to syphilis, be certain that syphilis was the cause of death (as opposed to dying with untreated syphilis where syphilis was not the cause of death.

Reviewer #3: The introduction revels lack of knowledge about syphilis. The data and its analysis do not support the conclusions. There is no way to check the quality of thes reported cases, no data about trends on use of public services and no prevalence data.

Reviewer #4: The topic is relevant to the public health area and presents important aspects for evidence-based practice by addressing a sexually transmitted infection in the elderly population. Despite the relevance and excellence in the preparation of the manuscript, it has weaknesses that put in check the uniqueness of the manuscript, since there are national publications on the subject with data from 2022 and the manuscript presents a temporal limitation with the use of data only until 2020. The study design is questionable, since modern epidemiologists such as Lash et al. Modern Epidemiology present scientific evidence for such aspects. However, the study also has a high descriptive potential, and the statistical analysis used constitutes only one of the first analyses of the composite possibility in the context of time series that are naturally defined as ecological study. Nevertheless, the categorization of the age range is also questionable, as there is no evidence of age up to 120 years, which raises the questioning of the quality of the data. The discussion presents relevant points, but little expressive regarding the importance of completeness and completion of official documents of notification of diseases that should be exalted in the limitations of the study. Another important point is that despite understanding that the data are of free access, it should be made clear that the research was not submitted for ethical review. Finally, despite the relevance of the theme and quality of the writing, I recommend that the authors make the necessary adjustments and submit the manuscript for publication in a public health section.

7. PLOS authors have the option to publish the peer review history of their article (what does this mean?). If published, this will include your full peer review and any attached files.

Reviewer #1: No

Reviewer #3: No

Reviewer #4: No

---

## [Author Response · Author response to Decision Letter 1]

19 Jul 2023

We submit a revised version of the manuscript that addresses the points raised during the review process.

We included a rebuttal letter that responds to each point raised by the academic editor and reviewers. We upload this letter as a separate file labeled 'Response to Reviewers'.

We send A marked-up copy of my manuscript that highlights changes made to the original version, labeled 'Revised Manuscript changes tracked with text highlights; and we send an unmarked version of revised paper without tracked changes, labeled 'Manuscript'.

We thank you for the guidelines sent and reinforce our desire to publish in a mega journal such as PlosOne.

---

## [Decision Letter · Decision Letter 2]

4 Oct 2023

PONE-D-23-02461R2ACQUIRED SYPHILIS IN OLDER PEOPLE IN BRAZIL FROM 2010-2020PLOS ONE

Dear Dr. Cunha,

Thank you for submitting your manuscript to PLOS ONE. After careful consideration, we feel that it has merit but does not fully meet PLOS ONE’s publication criteria as it currently stands. Therefore, we invite you to submit a revised version of the manuscript that addresses the points raised during the review process.

In addition to the comments made by the three reviewers please address the following points in the revised version of the manuscript:

**1. Writing Clarity:**

Please enhance writing clarity, refine sentence structure, and minimize the use of parentheses for a more fluid narrative. Additionally, ensure the use of bias-free language throughout your manuscript.

**2. Supplemental Files:**

Ensure that supplemental files are appropriately cited in the document and adhere to the correct naming conventions.Please exclude the list of figures as a supplemental file.The supplemental file "notification_acquired syphilis" is currently in Portuguese. Please also include an English translation for this file.The remaining supplemental files are currently named in Portuguese, and their content is not clearly specified. To align with the publication standards, it is advisable to rename these files in English and provide concise descriptions to elucidate their respective contents.

**3. Abstract:**

Please address the temporal discrepancy in the abstract, where main findings refer to data up to 2018 while the research scope extends to 2020.

**4. Introduction:**

It would be beneficial to include additional references when stating "Some studies highlight a high prevalence of these infections in individuals over 60 years, including those in extreme age groups"Consider substituting the figurative language "invisible" in the statement "There is structural ageism, making their sexuality invisible."

**5. Methods:**

Please include a link to the database when referencing the data source.Clarify the statement "It is essential to point out that not all information contained in the SINAN notification form is made public" particularly regarding the transparency of the data presented in this manuscript.Consider using "was obtained" instead of "came from" in the statement "The information regarding these variables came from the SINAN database."

**6. Results:**

Please provide clarification regarding the statistical significance of the observed difference mentioned in the statement "It was higher in males (34.74 cases per 100,000 individuals) than in females (18.46 cases per 100,000 individuals)."Consider changing the caption of Fig 2, reflecting the prevalence of acquired syphilis in older people among different regions of Brazil.Kindly note that Fig 3 contains words in Portuguese. Please ensure compliance with language standards.Provide an explanation for the use of '*' in some numbers in Table 1.

**7. Discussion:**

Please correct the capitalization of "Covid-19" for precision.Prioritize the interpretation of data before drawing comparisons with other studies. Additionally, always describe the statistical significance when stating that a particular statistical pattern is more or less pronounced.

We look forward to receiving your revised manuscript.

Kind regards,

Delfina Fernandes Hlashwayo, M.Sc.

Academic Editor

PLOS ONE

Reviewers' comments:

Reviewer's Responses to Questions

**Comments to the Author**

1. If the authors have adequately addressed your comments raised in a previous round of review and you feel that this manuscript is now acceptable for publication, you may indicate that here to bypass the “Comments to the Author” section, enter your conflict of interest statement in the “Confidential to Editor” section, and submit your "Accept" recommendation.

Reviewer #5: All comments have been addressed

Reviewer #6: (No Response)

Reviewer #7: (No Response)

2. Is the manuscript technically sound, and do the data support the conclusions?

Reviewer #5: Yes

Reviewer #6: Partly

Reviewer #7: Yes

3. Has the statistical analysis been performed appropriately and rigorously? 

Reviewer #5: Yes

Reviewer #6: N/A

Reviewer #7: Yes

4. Have the authors made all data underlying the findings in their manuscript fully available?

Reviewer #5: Yes

Reviewer #6: Yes

Reviewer #7: Yes

5. Is the manuscript presented in an intelligible fashion and written in standard English?

Reviewer #5: Yes

Reviewer #6: No

Reviewer #7: No

6. Review Comments to the Author

Reviewer #5: Reviewer's report

Title of the article

ACQUIRED SYPHILIS IN OLDER PEOPLE IN BRAZIL FROM 2010-2020

Reference number

PONE-D-23-02461R2

Comments:

Thank you for the opportunity to review your interesting manuscript. I enjoyed reading it as I feel it improved thanks to the addressing of other reviewers’ comments and suggestions. However, as a Clinician, I do agree with Reviewer #3 as several points of the introduction must be improved (see Major Compulsory Revisions). Moreover, I do suggest updating the references by referring to recently published major articles in Infectious Diseases as Peeling RW, et al. Lancet. 2023. doi: 10.1016/S0140-6736(22)02348-0 (see Major Compulsory Revisions). I believe that this manuscript would benefit from a publication in a Public Health Journal. All things considered, I think that this manuscript might be accepted after major revisions.

- Major Compulsory Revisions

Please, at least at the beginning of the text, try to correctly address the name of the pathogen which is Treponema pallidum pallidum (in italics).

The authors state that “the primary and secondary types are easy to diagnose”. I appreciate the numerous historical events cited by the authors: considering this, the authors should know that syphilis has been always considered “the great imitator” by clinicians. Syphilis is not an easy disease to diagnose. Not only for its various clinical features, but also for the extremely complex interpretation of non-treponemal titers [Marchese V, et al. J Clin Med. 2022. doi: 10.3390/jcm11247499]. Please, try to mitigate your statement, and try to expand the clinical presentations by referring to the extremely wide differential diagnosis [Tiecco G, et al. Pathogens. 202. doi: 10.3390/pathogens10111364].

Please, try to update epidemiologic data from all over the world by referring to recently published major articles in Infectious Diseases [Peeling RW, et al. Lancet. 2023. doi: 10.1016/S0140-6736(22)02348-0].

Quality of written English

English syntax is optimal.

Reviewer #6: This study summarized the rate of syphilis diagnosis reported to SINAN system in Brazil, with a focus on elders aged 60-120. The study showed that the detection rate is increasing in this population, with some differences in different regions. This information is valuable. However, the manuscript is not written concisely and sometimes hard to follow the main point. The conclusion in the abstract is directly supported by the result, while the conclusion from lines 437 to 446 are not supported by this study, more like for discussion. There is lack of the details how the case was defined/diagnosed, how cure/death was tracked, and how some detection rates were calculated. To understand the significance of increased detection rate in elders, it should be compared with the change of detection rate in younger adults. If the detection rate in younger adults was similarly or more increased, then the public health should remain focused on younger adults.

1. Introduction is very long and should be more focused. Paragraph 3 can be shortened to one sentence. Paragraph 4 is not relevant other than penicillin. Remove 84 to 85.

2. Give more introduction of SINAN, either in the introduction part or in Materials and Methods, such as who is doing the reporting, how it is reported and what is the reporting criteria? How to ensure all cases are reported consistently over the years. The study is performing the temporal trend analysis, that could be invalid if the reporting criteria has been changed over the years. As introduced between lines 99 and 100, the improvement of reporting may explain the increase in syphilis cases in Brazil. Is there data to describe the improvement of reporting, such as the potential population covered by the reporting system? This is useful to understand the increased detection rate, that is based on geographic census. As indicated by authors in the reply, the information of disease stage is not available, this should be specified in Materials and Methods.

3. Line 135-137, need more clarification to understand ‘therefore’ and ‘secondary data’? Is this a portion of data since 2010?

4. Line 140-141, unclear the criteria for the diagnosis of acquired syphilis.

5. Lin 146-147, unclear how the reporting system works and how to track the cure and death history.

6. Line 132-133 and Line 153-155 are both about detection rate calculation. Please combine. Line 157-158, not clear what ‘constant’ was used and why?

7. Line 182, need to specify the age group of older people to avoid confusion.

8. Line 184-185, not sure how 12.84 is derived? It is not implicated in fig 1 and does not fit with male and female results (34.74 and 18.46). Is it the journal style to use ‘34,74’ cases instead of ’34.74’? It is more confusing when reading the fig 1 and fig 5 with ‘,’ instead of ‘.’

9. In fig 1, what is first symptoms? What is difference between fig 1 and fig 5. There is some difference but very small. Need more introduction in Materials and Methods or in figure legends. In both fig 1 and fig 5, please include the rate change among younger adults. This will help the readers to understand the significance of increased detection rate among elders.

10. Line 194-200, calculation of % in every 100,000 older people is misleading, if it is not adjusted for % of each level of education in every 100,000 older people. In other words, calculate detection rate in every 100,000 illiterate older people, or every 100,000 older people with high school education… Such information is required for later discussion in line 357-362.

11. Line 202-203, not sure the meaning of ‘even higher than the national rate’. Certain regions are expected to have rates higher than national rate, if the rate is not same across different regions. Does it mean national rate in total population? Please specify. It is better to show the detection rate in total population or younger adults in each region.

12. Line 210-211 and fig 3, the comparison should include the younger adults from 18 to 60, ideally split to different age sub-groups. This will help to understand the significance of the detection rate among elders. Line 45-46 should have the same context to indicate that the mean age of 68.04 years is for adults aged 60-120.

13. Line 216-222 and fig 4, without the introduction of how disease was tracked and how cure or death was reported, it is difficult to understand this part, especially in the absence of disease stage information. Also need to relatively compare with the changes in younger adults in order to justify the significance of changes in elders.

14. Line 285-287, 298-305 should be moved to introduction or Materials and Methods.

15. Line 326-332, not sure whether the difference in white and black is significant in this study. Does not seem to be relevant.

16. Line 336-337, did not address previous reviewer comment, need proof, such as a reference. Or correlate with other cited references from line 339-345.

17. Line 346-350, please discuss why the cited reference is relevant or different with this study, and what is the implication.

18. Line 351-352, similar to reference 30, please compare with the detection rate in younger adults in order to discuss the significance of increased detection rate in elders.

Reviewer #7: Dear Editor,

Thanks for the opportunity to review the manuscript entitled Acquired Syphilis in Aged People in Brazil from 2010-2020. It presents important local data on syphilis among older people, which represent an emergent population at high risk of Syphilis.

My review is based on the R2 version.

My suggestions:

Abstract

Background: I suggest the authors bring the syphilis concern among older people.

Methodology: consider the appointments below.

Conclusion: The conclusion should be focused on your results on older people.

Introduction

Page 3, line 56- “…although it was discovered in the late 1400s [2]”.

The sentence and reference two need to be more appropriate. This is a secondary reference that discusses the origin of Syphilis, not the discovery.

Page 3, line 61- Please review reference 4 in the context of this sentence.

I suggest deleting lines 68 to 77 and 78 to 83. This information is unnecessary.

Material and Methods

This is an ecological study because the unit is “time”, despite the authors using public micro or macro data!

Line 137. Is there any case aged 120 years?

The authors should show the characteristics of Brazil to the readers: number of regions, number of inhabitants, number of people aged 60 years and over, human development index, etc. Also, they need to describe clearly how the Brazilian system defines acquired syphilis.

Page 6, line 114. Is there someone aged 120 years?

Please present the percentual of missing data for each variable.

Jointpoint Regression – please include the reference of the software.

Results

Table 1 presents data only for North, Northeast, and Mid-West Regions! In addition, there is no legend! Please review it.

Discussion

Line 292. It would be best if you were more precise. What do you mean about it? Underreporting of death and cure? Or underreporting of syphilis diagnosis?

Line 293. I suggest replacing “manifestation of the disease” with “notification of the disease.”

Line 298-310. I suggest moving this information to Material and Methods.

Line 311 to 317. If the decrease in syphilis cases is due to COVID-19 (and very likely is), then It should be desirable to evaluate the syphilis trend forward.

Lines 322 and 323 – “On the other hand, the increase in testing coverage and reorganization of health units did not stabilize the upward curve of the disease.”

It is noteworthy, according to your data, in 2019, before the COVID-19 Pandemic, there was a slight reduction in cases. This could be a trend-forward that was interrupted due to the pandemic.

Lines 336-337 –Please provide a reference.

Lines 339-345 – Bernstein et al. is not an appropriate reference. The discussion is about the higher syphilis frequency among men vs. women. Does it occur in Brazil? Why does it happen? I think you should also compare your data with other Brazilian studies.

Lines 347-350- I didn’t understand the references 28 and 19 in this context! Please review it.

Lines 351 – 356 – Please discuss your data about syphilis. Syphilis among older people!

Lines 371-375 – I think this paragraph is loose!

Lines 376 and 381 - Provide references. In Brazil? Please rewrite this paragraph more clearly.

Lines 384 to 389 – Despite the data's fragility, they showed a high cure rate for syphilis. Discuss it!

I suggest an extensive English review.

7. PLOS authors have the option to publish the peer review history of their article (what does this mean?). If published, this will include your full peer review and any attached files.

Reviewer #5: No

Reviewer #6: No

Reviewer #7: No

---

## [Author Response · Author response to Decision Letter 2]

15 Nov 2023

Dear all, we have responded to all the suggestions and proposals made by the reviewers. We are very grateful for the contributions provided.

---

## [Decision Letter · Decision Letter 3]

14 Dec 2023

ACQUIRED SYPHILIS IN OLDER PEOPLE IN BRAZIL FROM 2010-2020

PONE-D-23-02461R3

Dear Dr. Cunha,

We’re pleased to inform you that your manuscript has been judged scientifically suitable for publication and will be formally accepted for publication once it meets all outstanding technical requirements.

Kind regards,

Delfina Fernandes Hlashwayo, Ph.D.

Academic Editor

PLOS ONE

Additional Editor Comments (optional):

Please consider removing the "Notifiable Diseases Information System - Older people of 60 to 120 Years old.xlsx DOI 10.5281/zenodo.10086130." below the figure caption.

Reviewers' comments:

Reviewer's Responses to Questions

**Comments to the Author**

1. If the authors have adequately addressed your comments raised in a previous round of review and you feel that this manuscript is now acceptable for publication, you may indicate that here to bypass the “Comments to the Author” section, enter your conflict of interest statement in the “Confidential to Editor” section, and submit your "Accept" recommendation.

Reviewer #5: All comments have been addressed

Reviewer #7: All comments have been addressed

2. Is the manuscript technically sound, and do the data support the conclusions?

Reviewer #5: Yes

Reviewer #7: Yes

3. Has the statistical analysis been performed appropriately and rigorously? 

Reviewer #5: Yes

Reviewer #7: I Don't Know

4. Have the authors made all data underlying the findings in their manuscript fully available?

Reviewer #5: Yes

Reviewer #7: Yes

5. Is the manuscript presented in an intelligible fashion and written in standard English?

Reviewer #5: Yes

Reviewer #7: Yes

6. Review Comments to the Author

Reviewer #5: I feel the manuscript much improved after addressing all reviewers' comments. No issue was detected.

Reviewer #7: The authors responded to the previous suggestions, improving the quality of the manuscript. However, I still have some points, as described below:

I suggest that the authors delete the phrase “although it originated in the late 1400s and was discovered in 1905 by Schaudinn & Hoffmann [2,3]”. Because this information is not appropriate, nor is its reference. Furthermore, it did not add any important information.

I think the article is too long. I understand that the authors included several suggestions from the reviewers. However, they could be more concise. For example, lines 83-87 are unnecessary. The author previously presented global and American syphilis data where the study was conducted. I think this is enough.

In lines 90-98, the authors could only present general Brazilian data, not including regional data.

Line 226, I think the 120-year-old information is a typo. Please reconsider this information because maintaining it disqualifies your data.

Table 1. Describe the APC, AAPC, and CI in the legend.

The discussion could also be more objective and concise.

7. PLOS authors have the option to publish the peer review history of their article (what does this mean?). If published, this will include your full peer review and any attached files.

Reviewer #5: No

Reviewer #7: No

---

## [Editor Report · Acceptance letter]

16 Jun 2024

PONE-D-23-02461R3 

PLOS ONE

Dear Dr. Cunha, 

I'm pleased to inform you that your manuscript has been deemed suitable for publication in PLOS ONE. Congratulations! Your manuscript is now being handed over to our production team.

Kind regards, 

on behalf of

Dr. Delfina Fernandes Hlashwayo 

Academic Editor

PLOS ONE